# Characterization of the Relationship between the Chaperone and Lipid-Binding Functions of the 70-kDa Heat-Shock Protein, HspA1A

**DOI:** 10.3390/ijms21175995

**Published:** 2020-08-20

**Authors:** Larissa Smulders, Amanda J. Daniels, Caroline B. Plescia, Devon Berger, Robert V. Stahelin, Nikolas Nikolaidis

**Affiliations:** 1Department of Biological Science, Center for Applied Biotechnology Studies, and Center for Computational and Applied Mathematics, College of Natural Sciences and Mathematics, California State University Fullerton, Fullerton, CA 92834-6850, USA; l.smulders@csu.fullerton.edu (L.S.); ajdaniels@csu.fullerton.edu (A.J.D.); dberger8@csu.fullerton.edu (D.B.); 2Department of Medicinal Chemistry and Molecular Pharmacology and the Purdue University Center for Cancer Research, Purdue University, West Lafayette, IN 47907, USA; cplescia@purdue.edu (C.B.P.); rstaheli@purdue.edu (R.V.S.)

**Keywords:** chaperone function, heat-shock proteins, lipid binding, phosphatidylserine, protein refolding

## Abstract

HspA1A, a molecular chaperone, translocates to the plasma membrane (PM) of stressed and cancer cells. This translocation results in HspA1A’s cell-surface presentation, which renders tumors radiation insensitive. To specifically inhibit the lipid-driven HspA1A’s PM translocation and devise new therapeutics it is imperative to characterize the unknown HspA1A’s lipid-binding regions and determine the relationship between the chaperone and lipid-binding functions. To elucidate this relationship, we determined the effect of phosphatidylserine (PS)-binding on the secondary structure and chaperone functions of HspA1A. Circular dichroism revealed that binding to PS resulted in minimal modification on HspA1A’s secondary structure. Measuring the release of inorganic phosphate revealed that PS-binding had no effect on HspA1A’s ATPase activity. In contrast, PS-binding showed subtle but consistent increases in HspA1A’s refolding activities. Furthermore, using a Lysine-71-Alanine mutation (K71A; a null-ATPase mutant) of HspA1A we show that although K71A binds to PS with affinities similar to the wild-type (WT), the mutated protein associates with lipids three times faster and dissociates 300 times faster than the WT HspA1A. These observations suggest a two-step binding model including an initial interaction of HspA1A with lipids followed by a conformational change of the HspA1A-lipid complex, which accelerates the binding reaction. Together these findings strongly support the notion that the chaperone and lipid-binding activities of HspA1A are dependent but the regions mediating these functions do not overlap and provide the basis for future interventions to inhibit HspA1A’s PM-translocation in tumor cells, making them sensitive to radiation therapy.

## 1. Introduction

HspA1A is a stress-inducible seventy-kilodalton heat shock protein (Hsp70) [1]. This protein is a molecular chaperone that plays critical roles in protein homeostasis and survival of stressed and cancer cells. In order to function as a molecular chaperone, HspA1A interacts with short hydrophobic regions of misfolded proteins in an ATP dependent manner. As a typical Hsp70, HspA1A contains a nucleotide binding domain (NBD) connected to a substrate binding domain (SBD; β-strand subdomain) via a hydrophobic linker, followed by the C-terminus α-helical region [1,2]. The NBD binds and hydrolyzes ATP, whereas the SBD binds exposed hydrophobic regions of proteins [3,4]. The chaperone performs its function by alternating between its open and closed conformations. This change is regulated by ATP binding and hydrolysis at the NBD and the resulting allosteric conformation change of the protein [3,4]. 

In addition to its intracellular anti-apoptotic functions, HspA1A also localizes in the plasma membrane (PM) where it is presented on the surface of many different human tumor types resulting in radiation insensitive and metastatically aggressive tumors [5]. Thus, inhibiting HspA1A’s PM translocation would augment current therapeutics and detail new ones.

Inhibition of HspA1A’s PM localization requires prior knowledge of the molecular mechanism that drives this translocation. However, the mechanism that HspA1A uses to translocate to the PM is largely unknown, because the protein does not contain any known lipid-binding domains or other signals necessary for protein translocation to or embedding in the PM. Nevertheless, multiple reports demonstrated that HspA1A binds to particular anionic lipids and further revealed that HspA1A’s PM localization, anchorage, and surface presentation depends on its interaction with phosphatidylserine (PS) [6,7,8,9].

Therefore, inhibiting the binding of HspA1A to this lipid might stop the protein from localizing at the PM and subsequently HspA1A’s cell-surface presentation. However, the regions of HspA1A that mediate the interaction with PS, as well as the relationship between the lipid-binding and chaperone functions, are largely unknown. This knowledge will provide critical information about the unknown regions of HspA1A that bind to PS, thus mediating the protein’s PM localization, and allow us for future interventions to specifically target the lipid binding function of HspA1A as opposed to targeting the chaperone function, which has detrimental side and off-target effects.

To further characterize the interaction of HspA1A with PS and elucidate whether lipid-binding regulates its chaperone function, we determined the effect of lipids on the secondary structure and chaperone function of HspA1A using the wild-type and a mutated form of the protein that is unable to function as a molecular chaperone. 

## 2. Results

### 2.1. Rational of Lipids Used in the Present Study

We used two different lipids (phosphatidylcholine (PC) and PS) because HspA1A binds with high affinity to PS but shows very low binding to PC [8,9,10,11]. Thus, these experiments would discriminate whether the presence of any lipid or lipid-specific binding is responsible for any observed change in HspA1A’s chaperone activities. Moreover, PC is the most abundant lipid in mammalian cells [12] representing a lipid HspA1A would come in contact with during its cellular lifetime. Further, we used two different species of each lipid because it is known that HspA1A binds peripherally to liposomes composed of POPS (monounsaturated hydrocarbon chains), while it embeds in the lipid bilayer of liposomes composed of DPPS (desaturated hydrocarbon chains) [8,11]. Thus, these experiments would discriminate between the effects of peripheral binding or embedding on chaperone function.

### 2.2. HspA1A Lipid-Binding Properties

HspA1A binds to specific anionic lipids, including different species of PS [6,7,8,9,11]. Between the different PS species tested, HspA1A embeds exclusively in liposomal membranes containing DPPS, while binds peripherally to less saturated species, i.e., DOPS and POPS [8,11]. Furthermore, the binding of HspA1A to different PC species is relatively low and non-specific and most likely not physiologically relevant [6,7,8,9,11]. 

To ensure that the recombinant proteins used in the current study have similar lipid-binding characteristics as the ones described in the literature, we first used the lipid vesicle sedimentation (LVS) assay and quantified the binding of HspA1A to different species of PC and PS. In agreement with the published literature [6,7,8,9,11], these experiments revealed that all three protein batches used in the current study bind to PS species significantly higher than to any of the PC species used (Figure 1 and Appendix A). Furthermore, our results revealed that HspA1A binds to DPPS significantly more than to POPS (Figure 1B).

### 2.3. Effects of Lipids on the Secondary Structure of HspA1A

After verifying that the HspA1A recombinant proteins used in this study bind to lipids as expected, we sought to determine the effect of lipid binding on the secondary structure HspA1A. For this reason, we used the well-established Circular dichroism (CD) spectrometry technique [13]. In these experiments, we used single lipid liposomes (LUVs) composed of POPC (control), POPS (peripheral binding), and DPPS (embedding).

These experiments revealed that lipid binding caused only subtle changes in the secondary structure of HspA1A irrespective of which lipid was used (Figure 2). Specifically, the presence of POPC caused a small increase and a subtle loss at the β-strand and α-helical regions of the protein, respectively. Differently, the presence of POPS, which HspA1A binds to peripherally, resulted in a small decrease of the β-strand and a subtle increase of the α-helical regions of the protein. Lastly, the presence of DPPS liposomes, which HspA1A binds and embeds, resulted in a small increase of the β-strand regions and a small reduction of the α-helical regions, similarly to POPC. However, these subtle changes were not consistent between the biological replicates and thus not significant (Figure 2).

### 2.4. Effects of Lipids on the ATPase Activity of HspA1A

HspA1A’s binding to PS is mediated largely by the nucleotide-binding domain (NBD) of the protein [8,9]. Furthermore, binding to ATP, but not ATP hydrolysis, results in almost a complete loss of binding to PS, while this loss is not significant when the protein is pre-incubated with the lipid [8,9]. The latter finding suggests that although the conformational changes of HspA1A that are caused by ATP-binding are important for lipid binding, the regions responsible for nucleotide- and lipid-binding do not overlap. Based on these observations, we predicted that the presence and binding to lipids would change the ATPase activity of HspA1A.

To test this prediction and determine the effect of lipid binding to the ATPase function of HspA1A, we measured the release of inorganic phosphate at different time points in the presence or absence of liposomes (LUVs) composed of different lipids (Figure 3). These results revealed that the presence of lipids results in a significant increase of released inorganic phosphate (Figure 3). Specifically, both POPC and POPC:POPS, as well as DPPC and DPPC:DPPS liposomes, almost doubled the released phosphate. These results support the notion that the presence of lipids increases the ATP hydrolysis activity of HspA1A. However, this increase is induced at a similar level by either the control lipids (POPC, DPPC) or by the experimental lipids (POPS, DPPS) (Figure 3). Therefore, these findings strongly suggest that the specific binding and embedding of HspA1A to POPS and DPPS do not specifically affect the ATPase activity of the chaperone. 

### 2.5. Effects of Lipids on the Refolding Activity of HspA1A

As noted above, HspA1A bind to DPPS primarily via the NBD region of the protein. In contrast, the substrate-binding domain (SBD) does not bind strongly to this lipid [8]. However, different reports revealed that the SBD is the region that actually embeds in liposomes composed of DPPS [8,11], suggesting that full-length HspA1A is necessary for the complete spectrum of the observed interactions. Based on these observations, we predicted the presence of DPPS liposomes, in which HspA1A embeds, will affect the refolding activity of the chaperone. 

To test this prediction and determine the effect of lipid binding to the refolding activities of HspA1A, we measured the rate of refolding of chemically denatured beta-galactosidase in the presence or absence of liposomes (LUVs) composed of different lipids (Figure 4). The results of these experiments revealed only small alterations of the HspA1A’s refolding activities in the presence of lipids. Although the effect size of these changes is small, they seem to be specific, because BSA, the control protein used in this assay, did not show similar alterations (Figure 4B). Specifically, the presence of POPC, a control lipid that HspA1A binds non-specifically, did not significantly alter the total refolded enzyme. Differently, the presence of the other control lipid (DPPC), resulted in a small and consistent increase in the total refolded enzyme, although, the rate of refolding was not significantly different compared to the refolding observed in the absence of the lipid (Figure 4C). The presence of POPS or DPPS, lipids that HspA1A bind, both revealed subtle but consistent increases in both the total refolded enzyme and the refolding rate (Figure 4). These results suggest that binding to specific lipids slightly increases the refolding activity of HspA1A. However, the increase observed by both POPS and DPPS was very similar, suggesting that binding rather than embedding is the cause of increased refolding. 

### 2.6. Lipid Binding Properties of the K71A Mutation of HspA1A 

In combination, the results presented above suggest that the regions responsible for the chaperone and the lipid-binding functions of HspA1A do not overlap. To further evaluate this notion, we characterized the lipid binding properties of a well-studied mutation, the K71A, which abolishes ATP hydrolysis and has no refolding and chaperone activities [14,15,16]. There were two main reasons for using the well-studied K71A mutation [16]. First, we predicted that if the chaperone (ATP binding and hydrolysis as well as client protein binding) functional sites overlap with the lipid-binding sites, then using a null-ATPase and loss-of-function mutant should restrict lipid-binding. Second, we predicted that because K71A shows specific conformational differences as compared to the WT protein [16,17], it might affect how HspA1A binding to PS, thus aiding in the characterization of the putative binding sites. 

We first used the LVS method with a single concentration of both proteins and lipids and compared the observed binding between the wild-type (WT) and the K71A proteins (Figure 5). The results from this assay revealed that not only the K71A mutation does not abolish lipid-binding but it actually shows higher total binding as compared to the WT HspA1A (Figure 5). 

We then determined the effect of lipids on the secondary structure of the K71A mutation using CD spectrometry using the conditions and rational described in Section 2.3. These experiments revealed that the mutation does not cause major alterations on the secondary structure of the protein (Figure 6A) with the exception of a small decrease in the α-helical regions of the protein. The incubation of the K71A with POPS caused a consistent, between biological replicates, loss of the α-helical regions and a substantial but not consistent between replicates increase at the β-strand regions of the protein (Figure 6B). Differently, the presence of DPPS liposomes, in which HspA1A binds and embeds, resulted in a small decrease of the β-strand regions of the protein (Figure 6C).

The changes observed in total lipid-binding and on the secondary structure of the K71A mutant prompted us to further investigate the lipid-binding properties of the K71A mutation. Our aim was to characterize the kinetics of lipid binding interaction for both the WT and the K71A mutation using SPR (Figure 7 and Table 1). 

The results of these experiments revealed several major differences between the two proteins as well as specific properties of the lipid-binding of HspA1A (Figure 7 and Table 1). First, in both cases the best fit for the data was the Two state model, which suggests an initial interaction followed by a conformational change resulting in complex stabilization. Second, the WT has a much higher maximal binding capacity (*R*_max_) as compared to the K71A mutation. Third, the association rate constant (*k*_a1_) for the WT protein is three times lower than the value for the K71A mutation, suggesting that the latter associates much faster with the lipids. Fourth, the dissociation rate constant (*k*_d1_) for the WT protein is more than 300 times slower than the K71A protein, suggesting that the K71A-lipid complex decays much faster than the WT-lipid complexes. Fifth, the rate constant for conversion of initial to final complex (*k*_a2_) of the WT is 100 higher than the K71A, suggesting a much faster conformational change for the WT. Sixth, the rate constant for conversion of the final complex back to the initial (*k*_d2_) for the WT is approximately 250 times higher compared to the K71A, suggesting that the modified WT-lipid complex converts back at a much faster pace than the K71A-lipid modified complex. Seventh, the dissociation equilibrium constant (*K*_D_) of K71A is six times lower than the WT protein, suggesting higher overall binding affinity for K71A. 

## 3. Discussion

Recent research revealed that HspA1A translocates to the PM and is presented in the cell surface of multiple cancer types resulting in radiation insensitive and aggressive tumors. Many reports revealed that HspA1A is found in association with sphingomyelin/cholesterol rich lipid domains (lipid rafts) [5,18,19,20,21]. Therefore, several investigators aim to target the cell surface/PM-localized HspA1A as a novel anti-cancer treatment [5,22,23,24]. It is also fairly well established that HspA1A’s PM localization depends on its interaction with intracellular PS and other lipids, including globotriaoslyceramide, which is primarily found in lipid rafts [6,7,25]. However, how HspA1A binds to PS, GB3, or other PM lipids remains a mystery because this protein does not contain any known lipid-binding domains or other PM translocation signals. Thus, if we manage to inhibit HspA1A’s interaction with PS (or other lipids), we might be able to augment current therapies by stopping the protein from translocating to the PM while still maintaining its chaperone functions. 

In this report, our goal was to advance our understanding of the HspA1A regions that bind to PS by determining whether the regions responsible for the chaperone functions of HspA1A overlap with the lipid-binding regions. 

Our experiments verified that the proteins we used bind to lipids as expected based on the published literature [8,10,11] and provided an important new finding that HspA1A binds significantly higher to saturated PS (DPPS) as compared to mono-saturated PS (POPS). This finding suggests that in addition to the ability of HspA1A to preferentially embed to DPPS containing liposomes, the protein also has a higher affinity for saturated PS species. These observations might have important physiological implications because several well-controlled lipidomics studies revealed an increase in saturated lipid species in stressed and cancer cells [26]. In addition to the WT HspA1A proteins we also determined that K71A, an ATPase-defective HspA1A mutation, binds to POPS liposomes. We further evaluated this initial finding using SPR for both forms of proteins. The finding that K71A binds specifically to lipids indirectly supports the prediction [8] that lipids and nucleotides are not accommodated by the same protein sites, because if they did the expectation would be the loss of a positive amino acid in the region to result in loss of lipid-binding. Furthermore, the comparison between the binding characteristics of the WT and the K71A proteins, provided new information and basic clues on the interaction between HspA1A and POPS liposomes as well as the basis for explaining the effect of the K71A mutation on the lipid-binding properties and putative sites of the chaperone. Based on the results for both protein forms, the model that best describes the interaction of HspA1A with POPS liposomes assumes a two-step interaction. During the first step, HspA1A interacts with the liposomes. During the second step, the initial complex formed undergoes a conformational change, which accelerates the binding reaction. This Two state model is in agreement with the binding model proposed to explain the binding of HspA1A to DPPS, BMP, and phosphoinositides [10,27] using the LVS method. Furthermore, the proposed mechanism is in accordance with the well-established ability of HspA1A to change conformations after binding to nucleotides or other proteins (co-chaperones or client proteins) and allosterically communicate information between the two major domains of the protein. It is important to note here that under the conditions used HspA1A interacts peripherally with the POPS liposomes [8,11,28], thus the mechanism suggested above does not include the final step of this interaction, which is limited to saturated lipids, the protein embedding in the lipid bilayer [8,11]. Although this model is an attractive interpretation of the current literature, it can only serve as an additional indication of conformational changes of HspA1A upon binding to POPS, and warrants further investigation. 

In addition to the binding model, the comparison of the kinetics between the WT and K71A protein forms provide further information on the lipid-binding properties of HspA1A. Although the maximal binding capacity and the dissociation equilibrium constant are significantly different between the WT and K71A in agreement with the LVS assay, these values are less informative and the effect size of the change much less pronounced than the kinetics of the interaction. Analysis of the kinetics reveals that the K71A mutation results in a significantly faster association of HspA1A with POPS and very fast protein-lipid dissociation. In contrast, the K71A mutation results in significantly slower generation of the final HspA1A-POPS complex (the product of the presumed conformational change) and slower conversion back to the original complex. Taking into account that the lysine residue in position 71 is buried within the ATP binding cleft, a plausible explanation of these observations could be that the mutation slows the protein’s conformational changes due to the binding to lipids. This idea is indirectly supported by the fact that the K71A mutation results in a protein that does not undergo a conformational change or release peptide in the presence of ATP, although the protein still binds to the nucleotide [16,17]. 

Several reports have revealed that HspA1A embedding in the lipid bilayer results in oligomerization of the protein [11,18,19,20,21,29,30], which by changing the quaternary structure of the complex could further support the conformational change concept. Indeed, de-oligomerization of HspA1A results in impairing of clathrin-independent endocytosis and less protein at the cell surface [21]. To assess whether major changes in the secondary structure of HspA1A occur because of either binding to or embedding into the lipid bilayer we used CD spectrometry. Our results strongly suggest that under the conditions tested lipid binding and embedding do not cause major and specific alterations of the WT protein structure. This finding partially contrasts the consistent loss of the α-helical region when K71A protein was incubated with POPS liposomes. These findings agree with results obtained using bis-(monoacylglycerol)-phosphate (BMP; [28]) and strongly suggest that although lipids might alter the conformation of HspA1A [8] they do not seem to affect the overall protein structure. Furthermore, these conformational changes might be very small compared to the binding of nucleotides in the presence of Mg^2+^ [31] or transient and very fast. The latter notion is supported by the kinetics of the interaction generated with SPR, which suggest a very rapid turnover for the WT protein and considerably lower transition for the K71A mutation.

Our next set of experiments determined that the addition of liposomes results in a significant increase in ATP hydrolysis. However, all lipids used similarly induced these changes, irrespective of whether HspA1A binds them with high affinity or not. These observations suggest that binding to lipids or embedding in the lipid bilayer does not specifically affect the ATPase activity of HspA1A. These findings are in agreement with previous research showing that the addition of ATP lowers lipid binding to background levels, but this loss-of-binding is less pronounced when HspA1A and lipids are pre-incubated prior to the addition of ATP [8,9,27]. Similar results were observed for another molecular chaperone (Hsp90), which also binds to lipids, although this study did not assess whether this increase is the result of specific lipid-binding [32]. Another lipid, sulfatide (SGC), induces oligomerization of HspA1A by locking it a high-affinity state to unfolded proteins and blocking the binding to ATP [30]. However, it is not clear whether this was the result of specific binding or if this was due to the presence of the lipid. Together these observations allow us to speculate that the presence of lipids that HspA1A interacts with, even with low affinities (e.g., PC, the most abundant phospholipid in mammalian cells), might stabilize the protein thus resulting in higher ATP hydrolysis.

The results of the ATPase assay imply that although in structural proximity [13], the regions that mediate ATP binding and hydrolysis and lipid binding do not overlap. The finding that the K71A mutation, which renders the chaperone unable to hydrolyze ATP, binds to POPS with affinity comparable to the WT’s protein, further supports the idea that ATP hydrolysis and lipid-binding are distinct functions mediated by non-overlapping regions. 

Our last set of experiments determined that the addition of liposomes has small effects on the refolding activity of HspA1A. These changes seem to be specific and clearly relate the refolding rate of the chaperone to the binding to specific lipids (i.e., POPS and DPPS). Nevertheless, the effect size of these changes (less than one-fold maximal increase) is relatively small and might not be physiologically important. This notion further strengthens our prediction that the regions of the protein responsible for lipid binding and refolding although related do not overlap. This notion is further supported by the increase in refolding rate, because if the lipid and the protein client competed for the same sites, the expected result would be a decrease in the refolding rate. Another possibility could be that the presence of lipids may directly influence refolding of β-galactosidase as has been described for different membrane proteins [33,34,35]. Although this possibility cannot be formally excluded it seems less probable because most of the refolding increase observed was PS specific and the BSA controls did not show a similar increase. Based on our findings and the current literature, the changes in both the ATPase and refolding activities could be explained by the effect of the lipid in either stabilizing the chaperone, or causing conformational changes that favor ATP hydrolysis and protein refolding. Although attractive, these speculations warrant further experimentation. 

In conclusion, our results reveal that similarly to Hsp90, in which the membrane-deforming and chaperone functions are separated [36], the regions mediating the chaperone and lipid binding functions of HspA1A do not overlap. This knowledge provides new information towards elucidating the lipid-driven mechanism of HspA1A’s translocation to the surface of tumor cells. These mechanistic details will allow us to specifically target the lipid-binding function of HspA1A in order to inhibit the lipid-driven translocation of HspA1A to the cell surface of tumor cells, making them sensitive to radiation therapy.

## 4. Materials and Methods 

### 4.1. Lipids, Chemicals, and Reagents

All lipids were purchased from Avanti Polar Lipids, Inc. (Alabaster, AL, USA). The lipids used in this study were DPPC: 1,2-dipalmitoyl-sn-glycero-3-phosphocholine; DPPS: 1,2-dipalmitoyl-sn-glycero-3-phosphoserine (sodium salt); POPC: Phosphatidylcholine 1-palmitoyl-2-oleoyl-sn-glycero-3-phosphocholine]; and POPS: Phosphatidylserine 1-palmitoyl-2-oleoyl-sn-glycero-3-phospho-L-serine] (sodium salt). Other common chemicals and reagents, for example, antibiotics, buffers, and growth media, were obtained from Fisher Scientific or Sigma-Aldrich (St. Louis, MO, USA).

### 4.2. Generation of Recombinant Wild-Type and Mutated Clones, Proteins, and Protein Purification

The mouse cDNA clone containing the *hspA1A* gene sequence, accession number BC054782 was used to generate the recombinant clones used in this study. The K71A mutation was generated using site directed mutagenesis, as previously described [14,15]. The reaction conditions, primers, and pet22b+ constructs as well as the protein production and purification are described in detail in [8,10,15,37]. 

### 4.3. Generation of Liposomes

Two different sizes of liposomes were generated in this study. Small unilamellar vesicles (SUVs) were generated by the freezing and thawing method as described in [8,9,10,27] and were used for assessing the lipid-binding properties of HspA1A because they are pelleted uniformly. To assess the effect of lipids on the structure, as well as the ATP hydrolysis and refolding activities of HspA1A, large unilamellar vesicles (LUVs) were used because of their higher stability. LUVs were also used for the Surface Plasmon Resonance (SPR) assays, because of their uniformity in coating the L1 sensor chip surface and stability. LUVs were generated using the extrusion method as described by Avanti Polar lipids (https://avantilipids.com/divisions/equipment-products/mini-extruder-extrusion-technique). The lipid composition and ratios used are provided in the figures. In all experiments, we aimed to generate liposomes that simulate physiological ratios of lipids in a typical membrane (e.g., PC:PS ratio of 80:20 mol/mol; [12]).

### 4.4. Lipid Binding Assays

To quantify the lipid-binding of the different HspA1A (wild-type (WT) and K71A mutant) recombinant protein batches the liposomal vesicle sedimentation (LVS) assay [8,9,10,27] was used. The assay involves the quantitative pelleting of liposomal vesicles from a protein and lipid mixture. After centrifugation the amount of protein bound to the lipids was then calculated by comparing the amount of protein that remained in the supernatant to the amount of protein that was found in the liposomal pellet. The assay and the conditions used to assess the lipid-binding properties of HspA1A were performed as described in [8,9,10,27]. 

To characterize the kinetics of the binding of HspA1A to LUVs we used surface plasmon resonance (SPR) [37]. The SPR experiments were performed using a BIACORE™ X100 (Cytiva, Marlborough, MA, USA) equipped with a L1 sensor chip. All SPR measurements were performed at 25 °C. The coating of the L1 sensor chip was performed as described in [38]. Lipid vesicles containing either POPC (100) or POPC:POPS (80:20) were injected at 5 μL/min to give a response of 3000 response units (RU) for the control channel or the active surface channel, respectively. Following lipid coating three injections of 10 μL of 50 mM NaOH (at 100 μL/min) were used to stabilize each lipid layer. In all SPR experiments, 80 to 90 μL of protein (in 10 mM HEPES (pH 7.4), 100 mM NaCl) was injected at a flow rate of 5 μL/min, to reach the saturation of binding signal (*R*_eq_). The lipid surface was regenerated using three injections of 10 μL of 50 mM NaOH at 30 μL/min. After sensorgrams were obtained for five different concentrations of each protein each of the sensorgrams was corrected for the refractive index change by subtracting the control surface response value. The data were analyzed using the BIACORE™ X100 Evaluation software v2 (Cytiva, Marlborough, MA, USA). We initially aimed to fit the data to a simple 1:1 interaction model using a nonlinear least-squares analysis of the binding isotherm and the global data analysis option available within the software. However, because the nonlinear regression did not saturate for the 1:1 model, we used the Two state reaction model, which describes a 1:1 binding of protein (A) to immobilized lipids (B) resulting in the initial complex (AB) followed by a conformational change that stabilizes the complex (AB*). For simplicity, the model assumes that the transition from the initial complex (AB) to the final complex (AB*) and the reverse occur only when A and B are bound to each other. The kinetic parameters are: *k*_a1_: association rate constant for formation of the initial complex AB; *k*_d1_—dissociation rate constant for complex AB; *k*_a2_—rate constant for conversion of AB (initial complex) to AB* (modified complex: typically, a conformational change); *k*_d2_—rate constant for conversion of AB* to AB. Three batches of each protein were used and the data are presented in Table 1.

### 4.5. Circular Dichroism (CD) Spectra

To test the effects of lipids on the structure of recombinant WT and mutated HspA1A protein we used circular dichroism (CD) spectra analysis [13]. CD measurements were recorded on a Jasco J-810 Chirascan spectrophotometer (Applied Photophysics, Leatherhead, UK) with temperature controlled by a Peltier Type Control System PFD 425S. For these experiments, we used 1 mg/mL recombinant HspA1A protein in a 50 mM Phosphate Buffer (pH 7.4) at 25 °C. Measurements were conducted in a 50 µL 0.1-mm pathlength Hellma Suprasil^®^ Quartz cuvette (Sigma-Aldrich Catalogue Z805963). Spectra were recorded from 190 to 280 nm using a 0.1 nm step size with 0.5 s time-per-point (with adaptive sampling), with each spectrum representing an average of three accumulations. Data of buffer and liposome suspensions without protein were also measured and their values were subtracted from the final protein spectra as background. Data were analyzed and plotted using the CD Analysis and Plotting Tool [38]. 

### 4.6. ATP Hydrolysis Assay

To test the effects of lipids on the function of HSPA1A, we determined whether the addition of liposomes affects the ability of HSPA1A to hydrolyze ATP. The ATPase assay used is a colorimetric assay that measures the amount of free inorganic phosphate. The assays were performed as described in [10,14,15] with the sole difference the incubation of HspA1A (1 μM) with 1 mM of LUVs for 30 min at 30 °C before the addition of ATP and the initiation of the reaction. This pre-incubation was performed because previous research has established that simultaneous incubation of ATP and lipid with HspA1A results in significant loss of lipid binding. In contrast, when the lipid and the protein are pre-incubated and the ATP is added later, the lipid-binding remains largely unchanged. The phosphate released in three independent experiments was plotted as the change of phosphate release over time [10,14,15]. The K71A mutation is known to minimize hydrolysis of ATP to control levels [16] and we have also used as a negative chaperone control and characterized its properties in [14,15].

### 4.7. β-Galactosidase Refolding Assay

To test the effects of lipids on the refolding function of HSPA1A, we determined whether adding lipids alter the ability of HspA1A to refold denatured β-galactosidase [10,39]. To this end, β-galactosidase was chemically denatured and the recovery of the enzyme’s activity relative to the activity of the native enzyme was measured after the addition of ATP and recombinant HspA1A. BSA and reactions with no enzyme were used as negative controls. The assay was performed as described in [10] and as in the ATPase assay, HspA1A (1μM) was pre-incubated with 1 mM of LUVs for 30 min at 30 °C. The results were analyzed as described in [10] and plotted as percent of activity based on the activity of the native β-galactosidase. The refolding rate was calculated by using the linear regression line of all time points.

### 4.8. Statistical Tests

One-way ANOVA with post-hoc Tukey HSD Test was used to determine statistical significance. In all cases a *p* value <0.05 was considered statistically significant. The boxplots were generated using the R software (http://shiny.chemgrid.org/boxplotr/) [40].

## Figures and Tables

**Figure 1 ijms-21-05995-f001:**
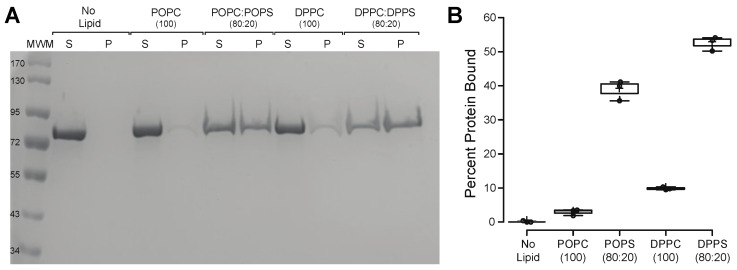
HspA1A binds to liposomes containing phosphatidylserine. A single concentration of liposomes (1 mM total lipid) containing different lipids in a specified molar ratio (see figure) was incubated with 1 μM of protein. After centrifugation of the liposome–protein mixture, the amount of lipid–protein binding was determined by comparing the protein that remained in the supernatant to the protein that was pelleted. (**A**) Representative SDS-PAGE gel electrophoresis of the two fractions (supernatant (S) and pellet (P)) after centrifugation of HspA1A and liposomes composed of different phosphatidylcholine (PC) and phosphatidylserine (PS) lipid species. These gels were overloaded to show protein purity; (**B**) The binding between HspA1A and lipids was quantified and the graphs are expressed as the percentage of protein bound to lipid vesicles (*Y*-axis). Error bars represent standard deviations for three independent experiments. The *p*-values for some of the comparisons were No lipid-POPC, *p* = 0.308756; No lipid-DPPC, *p* = 0.000308; POPS-DPPS, *p* = 0.000153.

**Figure 2 ijms-21-05995-f002:**
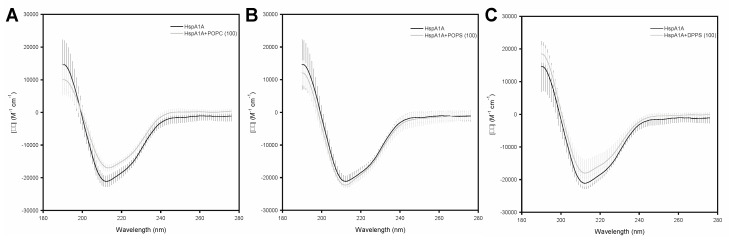
Circular dichroism (CD) experiments reveal subtle alterations of the secondary structure of HspA1A in the presence of different lipids. Mean residue CD extinction coefficient values (Δε in Μ^−1^·cm^−1^) were plotted and analyzed using the CD Analysis and Plotting Tool. Experiments were performed using three different protein batches, and the bars represent standard deviation (SD). The graphs compare the Δε values between HspA1A alone (black line) and HspA1A supplemented with (**A**) POPC, (**B**) POPS, and (**C**) DPPS liposomes.

**Figure 3 ijms-21-05995-f003:**
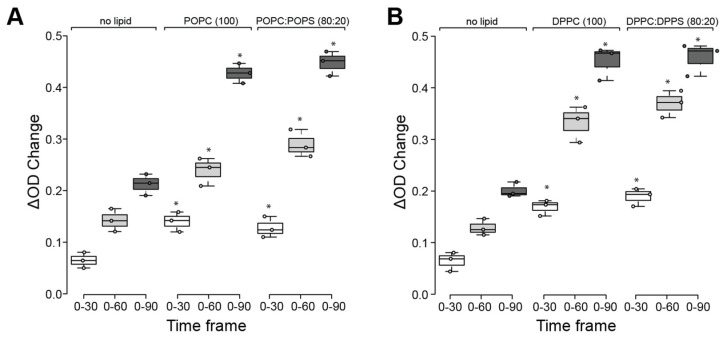
The presence of lipids, but not lipid-binding, increases the ATP hydrolysis activity of HspA1A. A colorimetric assay measuring the amount of free inorganic phosphate (Pi) was used. No HspA1A containing samples were used to control for spontaneous ATP hydrolysis, and the corresponding optical density (OD) values were subtracted from the samples’ values. The experiments were performed in the absence or presence of (**A**) POPC and POPC:POPS and (**B**) DPPC and DPPC:DPPS liposomes. The data are plotted as OD change (ΔOD) and are presented for three different periods (time frame). Each experiment was repeated three times. In all graphs, centerlines show the medians; box limits indicate the 25th and 75th percentiles as determined by R software; whiskers extend 1.5 times the interquartile range from the 25th and 75th percentiles. A *p*-value <0.05 is denoted with a *. The *p*-values of the statistical tests are provided in Appendix A.

**Figure 4 ijms-21-05995-f004:**
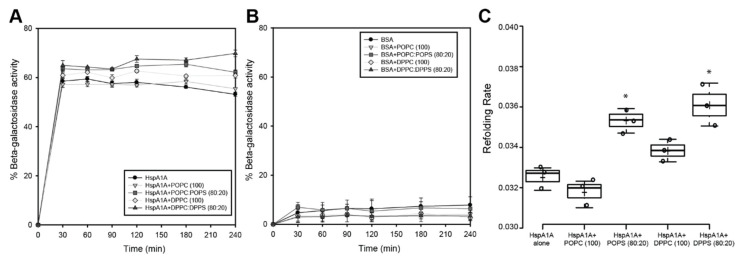
The binding to phosphatidylserine results in a small but consistent increase of the refolding activity of HspA1A. In these experiments, β-galactosidase was chemically denatured and the recovery of the enzyme’s activity was determined by hydrolysis of ortho-nitrophenyl-β-galactoside as compared to the activity of the native enzyme. (**A**). The data are plotted as percent of activity based on the activity of the native β-galactosidase; (**B**) BSA and reactions with no enzyme were used as negative controls; (**C**) The refolding rate was calculated by using the linear regression line of all time points. A *p*-value <0.05 is denoted with a *. The *p*-values of the statistical tests are provided in Appendix A.

**Figure 5 ijms-21-05995-f005:**
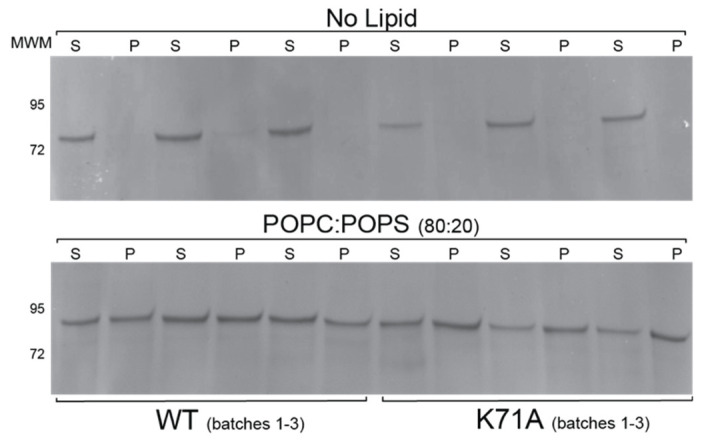
The K71A mutation binds to liposomes containing phosphatidylserine higher than the wild-type HspA1A. In these experiments, a single concentration of liposomes (1 mM total lipid) was used composed of POPC:POPS (80:20 mol:mol) incubated with 1 μM of protein. SDS-PAGE gel electrophoresis of the supernatant (S) and pellet (P) fractions of wild-type and K71A HspA1A without liposomes (Top; negative control) and with liposomes composed of POPC:POPS (80:20 mol:mol).

**Figure 6 ijms-21-05995-f006:**
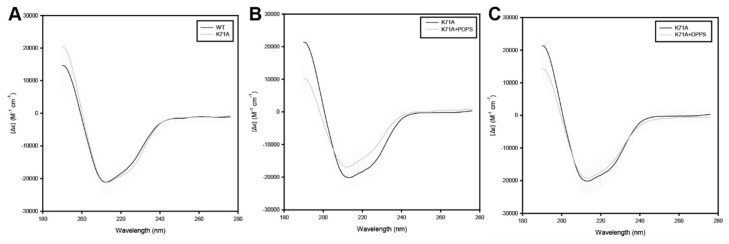
Circular dichroism (CD) experiments reveal subtle alterations of the secondary structure of the K71A mutated HspA1A in the presence of different lipids. Mean residue CD extinction coefficient values (Δε in Μ^−1^·cm^−1^) were plotted and analyzed using the CD Analysis and Plotting Tool. Experiments were performed using three different protein batches, and the bars represent SD. The graphs compare the Δε values between wild-type HspA1A and K71A without liposomes (**A**), and K71A HspA1A supplemented with (**B**) POPS, and (**C**) DPPS liposomes.

**Figure 7 ijms-21-05995-f007:**
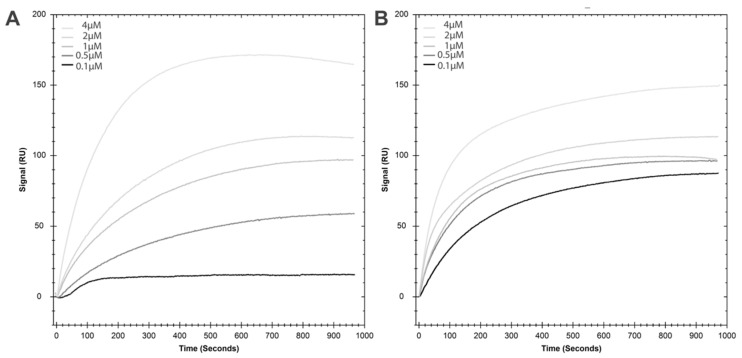
Surface plasmon resonance (SPR) binding measurements of wild-type (**A**) and K71A (**B**) HspA1A with PS-containing vesicles. Increasing concentrations of HspA1A were injected into the Biacore™ X100 instrument, where flow cell 1 contained POPC (100) vesicles and flow cell 2 contained POPC:POPS (80:20). Flow cell 1 and flow cell 2 response unit (RU) values before protein injections were normalized to zero to present HspA1A binding sensorgrams as shown.

**Table 1 ijms-21-05995-t001:** Kinetic parameters of the HspA1A (WT) and K71A mutant based on the SPR measurements.

**Protein**	**R_max_ (RU)**	**k_a1_ (1/Ms)**	**k_d1_ (1/s)**	**k_a2_ (1/s)**	**k_d2_ (1/s)**	**K_D_ (M)**
WT1	186.73	3.31 × 10^3^	1.61 × 10^−4^	3.48 × 10^−3^	2.81 × 10^−3^	1.22 × 10^−6^
WT2	176.98	2.84 × 10^3^	3.03 × 10^−4^	2.53 × 10^−3^	4.59 × 10^−3^	2.41 × 10^−6^
WT3	166.47	2.80 × 10^3^	1.26 × 10^−4^	2.13 × 10^−3^	5.47 × 10^−3^	1.60 × 10^−6^
K71A1	150.69	1.06 × 10^4^	2.67 × 10^−6^	2.23 × 10^−2^	2.14 × 10^−2^	2.94 × 10^−7^
K71A2	138.94	1.10 × 10^4^	8.81 × 10^−6^	1.48 × 10^−2^	1.57 × 10^−2^	2.47 × 10^−7^
K71A3	156.86	1.07 × 10^4^	6.32 × 10^−6^	1.69 × 10^−2^	1.73 × 10^−2^	2.86 × 10^−7^
Mean	**R_max_ (RU)**	**k_a1_ (1/Ms)**	**k_d1_ (1/s)**	**k_a2_ (1/s)**	**k_d2_ (1/s)**	**K_D_ (M)**
WT	176.73	2.99 × 10^3^	1.97 × 10^−4^	2.71 × 10^−3^	4.29 × 10^−3^	1.74 × 10^−6^
K71A	148.83	1.08 × 10^4^	5.94 × 10^−6^	1.80 × 10^−2^	1.81 × 10^−2^	2.75 × 10^−7^
SD	**R_max_ (RU)**	**k_a1_ (1/Ms)**	**k_d1_ (1/s)**	**k_a2_ (1/s)**	**k_d2_ (1/s)**	**K_D_ (M)**
WT	10.13	2.84 × 10^2^	9.38 × 10^−5^	6.88 × 10^−4^	1.35 × 10^−3^	6.04 × 10^−7^
K71A	9.1	1.97 × 10^2^	3.08 × 10^−6^	3.84 × 10^−4^	2.95 × 10^−3^	2.51 × 10^−8^
***t*-test (WT vs. K71A)**						
*p* value	0.0239	<0.0001	0.0222	0.0024	0.0018	0.0135

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
