# Peer review of "Characterization of the Relationship between the Chaperone and Lipid-Binding Functions of the 70-kDa Heat-Shock Protein, HspA1A"

_ijms, 2020, doi:10.3390/ijms21175995_

Round 1

Reviewer 1 Report

The authors aimed to investigate Hsp70 - PS interaction, which, they claim, drives the Hsp70 translocation and plasma membrane localization in stressed and cancer cells. The paper is clearly written and the evidences justify the conclusions. All the experimental protocols are given in sufficient detail.

Minor comments:

-         While PS-driven Hsp70 membrane binding is appealing other concepts of vesicular translocation and binding to e.g. sphingomyelin/cholesterol rich lipid domains extracellularily should be mentioned in the discussion (see references in Balogi et al review).

-         Lipids may directly influence refolding of proteins. This should be considered.

-        There are other studies that experimentally investigated Hsp70 oligomerization and its lipid binding (e.g Nimmervoll et al 2015). This should be mentioned and discussed in the light of the current data.

-       Conclusions should be carefully drawn as no cellular or in vivo data are provided.

Author Response

The authors aimed to investigate Hsp70 - PS interaction, which, they claim, drives the Hsp70 translocation and plasma membrane localization in stressed and cancer cells. The paper is clearly written and the evidences justify the conclusions. All the experimental protocols are given in sufficient detail.

Minor comments:

-         While PS-driven Hsp70 membrane binding is appealing other concepts of vesicular translocation and binding to e.g. sphingomyelin/cholesterol rich lipid domains extracellularily should be mentioned in the discussion (see references in Balogi et al review).

We added this concept in the discussion (please see lines XXX).

-         Lipids may directly influence refolding of proteins. This should be considered.

We discuss this notion in the revised manuscript (please see lines XXX).

-        There are other studies that experimentally investigated Hsp70 oligomerization and its lipid binding (e.g Nimmervoll et al 2015). This should be mentioned and discussed in the light of the current data.

We added this concept in the discussion (please see lines XXX).

-       Conclusions should be carefully drawn as no cellular or in vivo data are provided.

We carefully proofread the text to avoid any unsupported extrapolations. 

Reviewer 2 Report

The regions of HspA1A responsible for its interaction with lipids and how its lipid-binding activity relates to its chaperone functions are important questions but remains largely unknown. In this article, Smulders et al., aimed to characterize the unknown lipid-binding regions of HspA1A and to elucidate the relationship between its chaperone and lipid-binding functions. This work showed that one phospholipid, phosphatidylserine (PS) binding caused non-significant conformational changes of HspA1A secondary structure and had no effect on its ATPase activity. Interestingly, PS binding promotes HspA1A refolding activities. Additionally, a null-ATPase HspA1A mutant, HspA1AK71A alters the kinetics of PS binding with HspA1A. While the purposes of this study are of significance, some conclusions are lack of experimental supports, also the writing needs improvement.

Comments:

  1. Abstract needs improvements, some sentences are not clear. The readers would expect the abstract be as specific as possible. In line 27-28, “the kinetics of the binding are significantly different”, how different they are? In line 29, “these observations suggest a two-step binding model ….., but the latter half sentence did not describe this two-step model.
  2. In introduction, the authors should generally describe how HspA1A functions as a chaperone.
  3. In line 81, “In agreement with the published literature” the literature needs to be numbered there
  4. The last paragraph in the introduction section is more appropriate to be placed in the result 2.1 section. Additionally, the reasons why the ratio of 80:20 was chosen needs to be clear, why other ratio, e.g. 50:50 is not chosen?
  5. Line 120-124, the authors cited previous findings of “binding to ATP resulta in almost a …., while this loss is not significant when the protein is pre-incubated with the lipid [5,6] to lead their hypothesis that PS binding and nucleotide binding may not overlap. However, the findings they cited conversely support that PS binding and nucleotide binding may overlap, and indeed the citation [6] has shown that nucleotide binding domain binds to lipids.
  6. Figure 3, it would be better to add the statistical significance level, similarly to what the authors have done to Figure 4C.
  7. The authors aimed to uncover the relationship between regions of chaperone and lipid biding functions. While from Figure 7, the authors observed seven differences of lipid binding kinetics between WT and K71A mutated HspA1A, how these differences link to their conclusion of the two regions do not overlap? Using such mutant is a relatively indirect method, if it is possible to use some other methods to investigate this question, for example, by truncating proteins without known lipid-binding domain.

Author Response

Reviewer 2

The regions of HspA1A responsible for its interaction with lipids and how its lipid-binding activity relates to its chaperone functions are important questions but remains largely unknown. In this article, Smulders et al., aimed to characterize the unknown lipid-binding regions of HspA1A and to elucidate the relationship between its chaperone and lipid-binding functions. This work showed that one phospholipid, phosphatidylserine (PS) binding caused non-significant conformational changes of HspA1A secondary structure and had no effect on its ATPase activity. Interestingly, PS binding promotes HspA1A refolding activities. Additionally, a null-ATPase HspA1A mutant, HspA1AK71A alters the kinetics of PS binding with HspA1A. While the purposes of this study are of significance, some conclusions are lack of experimental supports, also the writing needs improvement.

Comments:

  1. Abstract needs improvements, some sentences are not clear. The readers would expect the abstract be as specific as possible. In line 27-28, “the kinetics of the binding are significantly different”, how different they are? In line 29, “these observations suggest a two-step binding model ….., but the latter half sentence did not describe this two-step model.

We revised the abstract and added the specific results (please see lines: XXX).

  1. In introduction, the authors should generally describe how HspA1A functions as a chaperone.

A short description was added to the first paragraph of the introduction (see lines: XXXX).

  1. In line 81, “In agreement with the published literature” the literature needs to be numbered there

The pertinent literature was added after this line (please see lines XXXXX).

  1. The last paragraph in the introduction section is more appropriate to be placed in the result 2.1 section.

We removed this paragraph from the introduction and included it as the first section of the results.

Additionally, the reasons why the ratio of 80:20 was chosen needs to be clear, why other ratio, e.g. 50:50 is not chosen?

We added a short explanation in the materials and methods section 4.3. Our goal was to mimic physiological phospholipid ratios (please see lines XXX and citation 12).

  1. Line 120-124, the authors cited previous findings of “binding to ATP results in almost a …., while this loss is not significant when the protein is pre-incubated with the lipid [5,6] to lead their hypothesis that PS binding and nucleotide binding may not overlap. However, the findings they cited conversely support that PS binding and nucleotide binding may overlap, and indeed the citation [6] has shown that nucleotide binding domain binds to lipids.

The cited literature supports the notion that the regions accommodating the nucleotide and lipid-binding functions do not overlap, because the order of addition of the nucleotide has a different effect on the lipid binding. If the two molecules competed for the exact same sites on the protein, then the order of addition should not make a difference. The fact that most of the lipid binding is accommodated by the nucleotide binding domain (NBD), does not necessarily mean that the sites overlap. The NBD is a relatively long molecule (approximately 380 amino acids) and binds ATP in the nucleotide binding pocket, which is buried inside the molecule. Although the lipid-binding sites are not yet known, we anticipate them to be closer to the surface of the molecule.

  1. Figure 3, it would be better to add the statistical significance level, similarly to what the authors have done to Figure 4C.

We added an asterisk to denote significance in the figure and modified the legend (please see XXXX).

  1. The authors aimed to uncover the relationship between regions of chaperone and lipid biding functions. While from Figure 7, the authors observed seven differences of lipid binding kinetics between WT and K71A mutated HspA1A, how these differences link to their conclusion of the two regions do not overlap? Using such mutant is a relatively indirect method, if it is possible to use some other methods to investigate this question, for example, by truncating proteins without known lipid-binding domain.

We agree with the reviewer that the K71A provides only indirect support for one of the goals of the manuscript, which is to investigate the relationship between the chaperone and the lipid-binding functions.

However, the findings indirectly support that the sites do not overlap, because if the two sites overlapped then the loss of a positive amino acid in the region would result in loss of lipid-binding. The results suggested otherwise, because the mutant binds to lipids with higher affinities than the WT protein.

The remaining data (i.e., kinetics of the interaction) revealed some mechanistic details of our second aim (identifying the sites responsible for lipid-binding).

We added in the results and discussion sections (please see lines XXX) a paragraph to clarify these issues.

Lastly, on the suggestion of using truncation mutants. In our previous manuscript, we investigated this idea using the NBD and SBD portions of the protein (see citation 8 in the main text). Although this research revealed that the interaction is mediated largely by the NBD region of HspA1A, it did not show the exact sites that bind to the lipids. Based on largely anecdotal evidence, the use of truncated mutation is discouraged for Hsp70s, because of their nature, structural characteristics, and the fact that any major truncation of the NBD results in insoluble protein. Furthermore, even if this approach was experimentally viable, it will again require the use of specific mutations to prove the point. Lastly, truncations would result in complete loss of the chaperone functions, thus it would not be feasible to delineate the relationship between the chaperone and lipid-binding functions.

Round 2

Reviewer 2 Report

The authors have addressed my comments carefully and the manuscript has been significantly improved. As a result, I now recommend the current version can be accepted for publication.